# The role of premorbid cognitive performance in the neuropsychological assessment of people with HIV in South Africa

Anna J. Dreyer[1,2]*, John A. Joska[2‡], Kevin G. F. Thomas[3‡], Caroline Sabin[4‡], Alan Winston[5,6‡], Saye Khoo[7‡], Sam Nightingale[8]

**1** Department of Clinical Neurosciences, University of Cambridge, Cambridge, United Kingdom, **2** HIV Mental Health Research Unit, Division of Neuropsychiatry, Department of Psychiatry and Mental Health Neuroscience Institute, University of Cape Town, Cape Town, South Africa, **3** Faculty of Humanities, University of Pretoria, Pretoria, South Africa, **4** Institute for Global Health, University College London, London, United Kingdom, **5** Imperial College Healthcare NHS Trust, London, United Kingdom, **6** Department of Infectious Disease, Imperial College London, London, United Kingdom, **7** Department of Pharmacology and Therapeutics, University of Liverpool, Liverpool, United Kingdom, **8** Departments of Psychiatry and Neurology, Neuroscience Institute, University of Cape Town, Cape Town, South Africa

☺ These authors contributed equally to this work.
‡ JAS, KGFT, CS, AW and SK also contributed equally to this work.
* anna.dreyer@uct.ac.za

## Abstract

### Background

Cognitive assessment in people with HIV may be confounded by psychosocial factors, especially those linked to childhood circumstances, which are rarely measured in studies investigating HIV effects on the brain. These factors may affect premorbid (pre-HIV) cognitive performance. We investigated how psychosocial factors influence premorbid cognitive performance in people with and without HIV.

### Methods

Participants were recruited into the CONNECT Study from a low-income area in Cape Town, South Africa. The Wechsler Adult Intelligence Scale (WAIS)-IV[SA] Information subtest served as an indicator of premorbid cognitive performance. We also measured global cognitive performance across seven cognitive domains (Global T-score), standard demographic variables (age, sex, education), and 12 psychosocial measures of childhood and adult circumstances, clustered using principal component analysis in prior analyses. Linear regression models examined associations between Global T-score and HIV status, adjusting sequentially for demographics and premorbid performance. Mediation analysis tested whether psychosocial effects on cognitive performance were mediated by premorbid performance.

which permits unrestricted use, distribution, and reproduction in any medium, provided the original author and source are credited.

**Data availability statement:** Our dataset contains highly sensitive information, including detailed psychosocial and socioeconomic variables from both childhood and adulthood, as well as HIV status, collected from patients attending a single clinic in Cape Town. This is a vulnerable population, and due to the granularity and combination of variables, there is a risk of indirect re-identification even after de-identification. Public deposition of these data would therefore compromise participant confidentiality and would not be consistent with the conditions under which ethical approval was obtained or with participants' informed consent. In line with PLOS policy, we are committed to data transparency while protecting participant privacy. The data are therefore available upon reasonable request to researchers who meet the criteria for access to confidential data, subject to appropriate data use agreements and, where necessary, institutional ethical approval. Data requests may be made to the following email address: hrec-enquiries@uct.ac.za.

**Funding:** Research reported in this publication was supported by the South African Medical Research Council, with funds received from the South African National Department of Health, UK Medical Research Council, and UK Government's Newton Fund. There was no additional external funding received for this study. The funders had no role in study design, data collection and analysis, decision to publish, or preparation of the manuscript.

**Competing interests:** The authors have declared that no competing interests exist.

## Results

177 people with HIV and 88 without HIV were assessed. People with HIV had lower premorbid cognitive performance than those without HIV (WAIS-IV[SA] Information: 3.89 vs. 5.24, p < .001). Premorbid performance was associated with 2/3 demographic, 5/7 childhood, and 2/5 adult psychosocial measures, as well as the *Childhood Psychosocial Variables* principal component (ps ≤ .002). Global T-scores were 3.95 points lower in people with HIV than those without (p < .001). Adjustment for demographics reduced this to 2.92 (p < .001), for premorbid performance to 2.70 (p < .001), and for both to 2.23 (p = .002). Mediation analysis indicated 39% partial mediation.

## Conclusions

In this setting, cognitive differences between people with and without HIV partly reflect premorbid disparities linked to childhood psychosocial circumstances. Analyses that do not account for these risks may misattribute low cognitive performance to HIV-associated brain injury. Both premorbid cognition and psychosocial history should be considered when interpreting adult cognitive assessments.

## Introduction

There is controversy around the prevalence of cognitive impairment in people with HIV, particularly when defined using criteria for the entity known as HIV-associated neurocognitive disorders (HAND) [1]. These criteria have been criticised for overestimating this prevalence, mainly due to a reliance on the quantitative results of cognitive performance testing without sufficient clinical context [1–5]. A recent meta-analysis of 123 studies that used these criteria (N = 35,513) estimated that, globally, 43% of people with HIV have HAND, with highest rates observed in Africa [6]. This estimate far exceeds rates observed in clinical practice and in research studies not using these criteria [7,8].

Cognitive test performance can be influenced by the psychosocial background of an individual, including education and socioeconomic status [9–11]. Our previous analysis from the Cognition, Neuropsychiatric Symptoms and Neuroinflammation Switching to Dolutegravir in Cape Town (CONNECT) study demonstrated that, in a cohort of South African people with HIV, lifetime psychosocial factors significantly affected cognitive performance during adulthood [12]. People with HIV reported experiencing greater levels of poverty than those without, with poverty also being linked to an increased risk of HIV acquisition in this geographical setting [13,14]. This difference in background levels of poverty created a bias to testing of cognitive performance which was incompletely mitigated by adjusting for the standard demographic variables typically controlled for in neuro-HIV studies (i.e., age, sex, ethnicity and years of education). The strongest associations with cognitive performance were psychosocial factors in childhood (e.g., quality of education, primary caregiver

occupation and highest level of education, and household assets). This suggests that psychosocial confounds to cognitive testing exert their influence early, before the person acquired HIV (vertical transmission was an exclusion in CONNECT).

In clinical practice, neuropsychologists aim to interpret current cognitive performance in the context of estimated premorbid functioning, i.e., the cognitive performance that would be expected if the person had not been exposed to the pathology in question (in this case, HIV acquisition). Premorbid functioning can be estimated based on a person's psycho-social background, such as educational and professional attainment, or quantified using tests of crystallized intelligence.

Tests assessing crystallized intelligence (e.g., the Wechsler Adult Intelligence Scale (WAIS) Information and Vocabulary subtests), rather than tests of fluid intelligence (e.g., the WAIS Digit Span), are considered a reasonable representation of premorbid cognitive abilities. Crystallized intelligence, which relies on accumulated information and is strongly associated with level of formal educational attainment, tends to remain stable (or even improve) with age, whereas fluid intelligence, which captures the ability to solve new problems, reason abstractly, and adapt to novel situations without relying on prior knowledge, tends to decline with age. Tests of crystalized intelligence are considered "hold" tests because they are less likely to be affected by brain injury than those assessing fluid intelligence, with WAIS Vocabulary and Information subsets thought to represent the best "hold" subtests [15–19].

Building on our previous findings that childhood psychosocial adversity contributes to observed cognitive differences between people with and without HIV, the present study introduces an estimate of premorbid cognitive performance to further clarify these associations. In this study, we estimated premorbid cognitive performance in a group of people with and without HIV to explore its association with current cognitive performance and psychosocial variables in childhood and adulthood. We hypothesised that lower premorbid cognitive performance would relate to more adverse psychosocial indices in people with HIV, particularly in childhood. We also sought to explore whether adjusting for premorbid cognitive performance mediates the effects of psychosocial variables on cognitive performance.

## Method

### Participants and setting

Participants ($N = 273$) were recruited as part of the CONNECT study, between Aug 12, 2019, and Sept 16, 2022, in the Gugulethu periurban area of Cape Town, South Africa [7,12]. Gugulethu was established in the 1960s as a residential area for Black South Africans; during the apartheid era, they were not permitted to live in the city centre. Recent census data indicate that most residents of Gugulethu are isiXhosa-speaking Black Africans of low socioeconomic status [20].

People with HIV ($n = 178$) were recruited from an HIV clinic in Gugulethu. Eligibility criteria required that they (a) were aged between 18 and 55 years, (b) had been receiving efavirenz-based antiretroviral therapy for at least 1 year, and (c) had been identified by the clinic as suitable candidates for the switch to dolutegravir-based antiretroviral therapy as per the South African national HIV treatment guidelines (i.e., HIV RNA < 50 copies/ml, or two samples <1000 copies/ml taken 3 months apart with evaluation and adherence support) [21].

People without HIV ($n = 95$) were also recruited from Gugulethu. They were matched to people with HIV by sex and age (within 5 years). To recruit people without HIV with as similar a set of socioeconomic indices as possible to those with HIV, we recruited friends, relatives, and associates of people attending the Gugulethu HIV clinic. Rapid test confirmed negative HIV status in these participants.

We excluded from participation all individuals with characteristics that could potentially confound their cognitive test performance: current substance misuse (Drug Use Disorders Identification Test for men > 5 or women >1); high-risk or harmful level alcohol use (Alcohol Use Disorders Identification Test >15) [22]; history of CNS infection or major head injury (loss of consciousness >30 minutes); uncontrolled neurologic conditions (e.g., seizure disorders, established cerebro-vascular disease); history of learning difficulty or severe intellectual disability; < 7 years total education; history of severe mental health disorder (schizophrenia, psychosis, or bipolar disorder); or vertical HIV acquisition for those with HIV. We also excluded individuals who at the time of enrolment were (a) being investigated or treated for active intercurrent illness

such as infection or carcinoma, (b) currently receiving treatment for tuberculosis, (c) known or suspected to be pregnant, or (d) not fluent in isiXhosa or English.

All measures were translated from their original English into isiXhosa using a standard forward–backward translation procedure to ensure conceptual and linguistic equivalence and were administered by a bilingual research assistant in the participant's language of choice.

## Ethics statement

All the participants provided informed verbal and written consent to study participation, including consent to use data from their medical records. The study was approved by University of Cape Town Faculty of Health Sciences Human Research Ethics Committee (017/2019).

## Estimated premorbid cognitive performance

The WAIS-IV<sup>SA</sup> Information subtest (i.e., the standard subtest adapted for use in South Africa) was used to estimate premorbid cognitive performance [23].

## Cognitive tests and outcomes

We used a standard 2-hour neuropsychological test battery to assess performance in the cognitive domains of executive functioning, verbal learning and memory, visuospatial learning and memory, verbal fluency, attention/working memory, information processing speed, and motor skills [24,25]. Tests were administered and scored by a bilingual-speaking neuropsychological technician supervised by a registered clinical neuropsychologist (AJD). More details of this test battery are described elsewhere [7].

For each individual test, raw scores were standardized to $z$-scores using the data from the group of people without HIV. The $z$-scores were then converted to $T$-scores ($M = 50$, $SD = 10$). Domain $T$-scores were calculated by taking the average of $T$-scores of the cognitive outcomes within each domain. A Global T-score was calculated by taking the average across domain $T$-scores.

## Psychosocial measures

We gathered self-report data regarding 12 psychosocial variables, 7 relating to the participant's childhood (ages 8–10 years unless otherwise stated) environment and circumstances and 5 relating to their current environment and circumstances.

**Childhood psychosocial variables.** *Primary and secondary school quality of education* was measured by recording the name and province of the primary school (Grades 0–7, 6–13 years of age) and secondary school (Grades 8–12, 13–18 years of age) the participant had attended. If the participant reported attending more than one primary or secondary school, data were recorded for the school of longest time enrolment. We classified each school using the South African government's national quintile system 2020 rankings based on the socioeconomic status of the school, where quintile 1 schools are of the lowest such status and quintile 5 schools the highest [26]. *Primary caregiver's highest level of education* was recorded as one of 6 categories, ranging from 'no formal education' to 'tertiary education.' The primary caregiver was defined as the biological mother; if she was not present, then the father; and then a guardian. *Primary caregiver's occupation category* was recorded as one of the 9 categories on the Hollingshead occupational scale (see Table 1 for list) [27]. A value for the *childhood asset index* was ascertained by asking the participant to record whether during childhood they had various items in working order in their household (refrigerator or freezer, vacuum cleaner or polisher, television, Hi-fi or music centre [radio excluded], microwave oven, washing machine, video cassette recorder or DVD player, running water, domestic worker, at least one car, flush toilet, built-in kitchen sink, an electric stove or hotplate and working

**Table 1. Standard demographic variables, psychosocial variables, and estimated premorbid cognitive performance in people with and without HIV.**

| Variables | People with HIV (*n*=177) M (*SD*) or n (%) as appropriate | People without HIV (*n*=88) M (*SD*) or n (%) as appropriate | p-value |
|---|---|---|---|
| **Estimate of premorbid cognitive performance** | | | |
| WAIS-IV[SA] Information subtest | 3.89 (1.87) | 5.24 (2.88) | <.001[a] |
| **Standard demographic variables** | | | |
| Sex (male) | 36 (20.34%) | 22 (25.00%) | .480 |
| Age at enrolment (years) | 40.7 (7.53) | 39.9 (9.03) | .489 |
| Years of education | 10.7 (1.26) | 11.3 (1.22) | <.001[a] |
| **Childhood psychosocial variables** | | | |
| Childhood asset index total score[c] | 4.12 (4.08) | 6.48 (4.13) | <.001[a] |
| Primary school quality of education quintile[d] | | | .0027[b] |
| Quintile 1 | 27 (19.29%) | 9 (11.39%) | |
| Quintile 2 | 30 (21.43%) | 18 (22.78%) | |
| Quintile 3 | 79 (56.43%) | 38 (48.10%) | |
| Quintile 4 | 3 (2.14%) | 11 (13.92%) | |
| Quintile 5 | 1 (0.71%) | 3 (3.80%) | |
| Secondary school quality of education quintile[e] | | | .0092 |
| Quintile 1 | 10 (6.80%) | 6 (7.79%) | |
| Quintile 2 | 42 (28.57%) | 13 (16.88%) | |
| Quintile 3 | 73 (49.66%) | 34 (44.16%) | |
| Quintile 4 | 16 (10.88%) | 11 (14.29%) | |
| Quintile 5 | 6 (4.08%) | 13 (16.88%) | |
| Primary caregiver highest level of education[f] | | | .0062[b] |
| 0 years | 23 (13.37%) | 3 (3.09%) | |
| 1–6 years | 40 (23.26%) | 17 (22.08%) | |
| 7 years | 24 (13.95%) | 5 (6.49%) | |
| 8–11 years | 71 (41.28%) | 35 (45.45%) | |
| 12 years | 9 (5.23%) | 9 (11.69%) | |
| >13 years | 5 (2.91%) | 8 (10.39%) | |
| Primary caregiver occupation category | | | .008[b] |
| Higher executives, major professionals, owners of large businesses | 0 (0%) | 0 (0%) | |
| Business managers of medium sized businesses, lesser professions | 6 (3.39%) | 8 (9.09%) | |
| Administrative personnel, managers, minor professionals, owners of small businesses | 2 (1.13%) | 2 (2.27%) | |
| Clerical and sales, technicians, small businesses | 4 (2.26%) | 7 (7.95%) | |
| Skilled manual | 7 (3.95%) | 4 (4.55%) | |
| Semi-skilled | 28 (15.82%) | 17 (19.32%) | |
| Unskilled | 38 (21.47%) | 24 (27.27%) | |
| Homemaker | 51 (28.81%) | 18 (20.45%) | |
| Student, disabled, no occupation | 41 (23.16%) | 8 (9.09%) | |
| Childhood trauma (CTQ score) | 36.46 (12.83%) | 35.27 (11.92%) | .178[a] |
| Exposure to violence (SECTV score) | 10.34 (7.88%) | 13.67 (7.94%) | .00082[a] |
| **Current psychosocial** | | | |
| Occupation category | | | .526[b] |
| Higher executives, major professionals, owners of large businesses | 0 (0%) | 0 (0%) | |

*(Continued)*

**Table 1.** (Continued)

| Variables | People with HIV (*n* = 177) *M* (*SD*) or n (%) as appropriate | People without HIV (*n* = 88) *M* (*SD*) or n (%) as appropriate | p-value |
|---|---|---|---|
| Business managers of medium sized businesses, lesser professions | 2 (1.13%) | 0 (0%) | |
| Administrative personnel, managers, minor professionals, owners of small businesses | 4 (2.26%) | 0 (0%) | |
| Clerical and sales, technicians, small businesses | 9 (5.08%) | 7 (7.95%) | |
| Skilled manual | 12 (6.78%) | 3 (3.42%) | |
| Semi-skilled | 28 (15.82%) | 12 (13.64%) | |
| Unskilled | 31 (17.51%) | 14 (15.91%) | |
| Homemaker | 8 (4.52%) | 2 (2.27%) | |
| Student, disabled, no occupation | 83 (46.89%) | 50 (56.82%) | |
| Depressive symptoms (CES-D > 16) | 17 (9.6%) | 20 (22.73%) | .007 |
| Monthly income from all sources (ZAR) | | | |
| R0-R499 | 29 (16.38%) | 13 (14.77%) | .566[b] |
| R500-R999 | 17 (9.60%) | 13 (14.77%) | |
| R1000-R1999 | 41 (23.16%) | 20 (22.73%) | |
| R2000-R2999 | 18 (10.17%) | 12 (13.64%) | |
| R3000-R3999 | 20 (11.30%) | 10 (11.35%) | |
| R4000-R4999 | 22 (12.43%) | 5 (5.68%) | |
| R5000-R9999 | 27 (15.25%) | 12 (13.64%) | |
| R10000-R19000 | 3 (1.69%) | 3 (3.41%) | |
| >R20000 | 0 (0%) | 0 (0%) | |
| Current asset index (per 1 point higher) | 7.68 (2.32) | 8.68 (2.02) | .00036[a] |
| Accommodation type | | | <.001 |
| Shack | 66 (37.29%) | 11 (12.5%) | |
| Wendy house/backyard dwelling | 52 (29.38%) | 18 (20.45%) | |
| Own/family house | 59 (33.33%) | 59 (67.05%) | |

*Note*. [a]Wilcoxon Rank Sum tests not t-test. [b]Fisher's Exact Test instead of Chi-square. [c]Subset of participants completed the measure (People with HIV: n = 176; People without HIV: n = 86); [d]Subset of participants completed the measure/school quintile not found for school (People with HIV: n = 140; People without HIV: n = 79); [e]Subset of participants completed the measure/school quintile not found for school (People with HIV: n = 147; People without HIV: n = 77); [f]Subset of participants completed measure (People with HIV: n = 172; People without HIV: n = 77); CTQ = Childhood Trauma Questionnaire; SECTV = Survey of Exposure to Community Violence; WAIS-IV[SA] = Wechsler Adult Intelligence Scale-IV for South Africa; WASI = Wechsler Abbreviated Scale of Intelligence

telephone); a point was given for each asset, leading to a possible total score of 14. *Childhood trauma* was measured using the Childhood Trauma Questionnaire (CTQ) recalled up to the age of 16 years [28]. *Exposure to violence* was measured using the Survey of Exposure to Community Violence (SECTV). This questionnaire asks respondents to indicate how often they saw or heard violent things in their home, neighbourhood, or school (not on television or in movies) when younger than 18 years of age.

**Current psychosocial variables.** *Monthly income* from all sources before tax was classified as being in one of nine brackets, spanning ZAR0 −>ZAR20,000. *Occupation category* was recorded as one of the nine categories on the Hollingshead occupational scale (see Table 1 for list) [27]. A value for the *current asset index* was ascertained by asking the participant to record whether they had various items in working order in their household (same list as above for *childhood*

*asset index*); a point was given for each asset, leading to a possible total score of 14. *Accommodation type* was measured by asking the participant what type of housing they lived in: none, 'shack' (a small informal house constructed from corrugated iron sheets attached to a basic wooden frame), 'wendyhouse or backyard dwelling' (a wendyhouse is a small timber cabin), or 'own or family house' (usually a brick house). *Depressive symptomatology* was measured using the Center for Epidemiological Studies-Depression (CES-D), with a standard cutoff of ≥16 used to indicate the presence of depressive symptoms [29].

**Principal component analysis (PCA) components.** In a previous analysis (published [12]), we used PCA on the continuous psychosocial variables to identify underlying patterns and reduce the dimensionality of the dataset for subsequent multivariable modelling. The PCA was conducted on the following variables: *CTQ score, SECTV score, primary caregiver occupation category, childhood asset index, primary school quality of education quintile, secondary school quality of education quintile, primary caregiver level of education, current asset index, monthly income*, and *occupation category*. The PCA revealed three components, labelled as *Childhood Psychosocial Variables, Current Psychosocial Variables* and *Experience of Childhood Trauma*.

## Statistical analyses

We used R Studio (R version 4.4.3) to complete all analyses.

First, to investigate magnitude of differences between the groups of people with and without HIV, we used *t*-tests (or Wilcoxon Rank Sum tests with continuity correction if variables were not normally distributed according to Shapiro-Wilk tests) and chi-square analyses with Yates' continuity correction.

Second, we used linear regression models to investigate univariable associations between demographic and psychosocial variables, PCA components and estimated premorbid cognitive performance.

Third, we used a linear regression model to investigate the univariable HIV association on global cognitive performance. We then built three multivariable linear regression models to estimate how much the inclusion of standard demographic variables, and estimated premorbid cognitive performance, attenuated the observed HIV association. The first model adjusted for standard demographic variables; the second adjusted for premorbid cognitive performance; the third adjusted for both standard demographic variables and premorbid cognitive performance.

Finally, we examined whether premorbid cognitive performance mediates the relationship between psychosocial variables and current global cognitive performance. Using linear regression models, we: (1) tested the association between the WAIS-IV[SA] Information subtest and Global T-score; (2) confirmed that the *Childhood Psychosocial Variables PCA component* was significantly associated with Global T-score (a finding previously reported in Dreyer et al. [12]); and (3) conducted mediation analyses including both the WAIS-IV[SA] Information subtest and the Childhood Psychosocial Variables component as predictors, with Global T-score as the outcome.

## Results

Data from the WAIS-IV[SA] Information subtest were incompletely collected in 1 person with HIV and 7 people without HIV; hence, all data from these participants were excluded from further analysis.

Table 1 shows that, on average, the sample of people with HIV (*n* = 177) scored significantly lower than the sample of people without HIV (*n* = 88) on the WAIS-IV[SA] Information subtest (3.89 vs. 5.24, p < .001).

Regarding standard demographic variables, on average people with HIV had completed significantly fewer years of education than people without HIV; analyses detected no other significant between-group differences (Table 1).

Regarding childhood psychosocial variables, on average people with HIV had significantly fewer assets, a significantly lower quality of education in both primary and secondary school, a primary caregiver with significantly less education and of a significantly lower occupational category, and significantly less exposure to violence. Analyses detected no significant between-group differences with regard to the experience of childhood trauma (Table 1).

Regarding current psychosocial variables, on average people with HIV were more likely to live in a shack and to have fewer assets. People without HIV had higher rates of depression. Analyses detected no significant between-group differences in occupation or income (Table 1).

The patterns of these between-group differences are the same as those reported previously for a larger sample of CONNECT participants [12].

### Associations of demographic and psychosocial variables with estimated premorbid cognitive performance in people with and without HIV

Table 2 presents results from the set of univariable association analyses. For each model, the outcome variable was the score on the WAIS-IV$^{SA}$ Information subtest (i.e., our estimate of premorbid cognitive performance).

With regard to standard demographic variables, sex and years of education, but not age, were significantly associated with the outcome. With regard to childhood psychosocial variables, all predictors except childhood trauma and exposure to violence were significantly associated with the outcome. With regard to current psychosocial variables, only assets and living in a family house vs. a shack were significantly associated with the outcome. This pattern was also reflected in the PCA components, where only the *Childhood Psychosocial Variables* component was significantly associated with the outcome.

### The effect of HIV infection on current cognitive test performance when controlling for standard demographic variables and premorbid cognitive performance

Table 3 presents results from the set of multivariable linear regression models. For each model, the outcome variable was current Global T-score (i.e., our estimate of current global cognitive performance).

In the unadjusted model, living with HIV was associated with a significantly lower current Global T-score: People with HIV had a Global T-score of 3.95 points less than people without HIV. The model adjusting for standard demographic variables reduced this score difference by 26%, to 2.92 points. The model adjusting for premorbid cognitive performance reduced the score difference even more, by 32%, reduction to 2.70. Finally, the model adjusting for both standard demographic variables and premorbid cognitive performance simultaneously reduced the HIV effect by 44%, to 2.23 points.

### Mediation analysis: Relationship between WAIS-IV$^{SA}$ Information, psychosocial variables and global cognitive performance

In this sample, the WAIS-IV$^{SA}$ Information subtest was significantly associated with Global *T*-score (Beta = 1.07, 95% CI = 0.77–1.37, p < .001). The *Childhood Psychosocial Variables* component also significantly associated with Global *T*-score (Beta = 1.83, 95% CI = 1.11–2.55, p < .001). However, the other two components (*Current Psychosocial Variables* and *Experience of Childhood Trauma*) were not significantly associated with the outcome. Note that these results were published in Dreyer et al. [12].

In the multivariate model with both WAIS-IV$^{SA}$ Information subtest score and the *Childhood Psychosocial Variables* component as predictors and Global *T*-score as outcome, WAIS-IV$^{SA}$ Information subtest score partially mediated the relationship between the *Childhood Psychosocial Variables* component and Global *T*-score (Fig 1). Specifically, in the unadjusted model, participants with worse childhood psychosocial circumstances had a Global *T*-score of 1.83 points lower than those with better such circumstances (Beta = 1.83, 95% CI = 1.11–2.55, p < .001). The subsequent model adjusting for WAIS-IV$^{SA}$ Information subtest score (Beta = 0.91, 95% CI = 0.60–1.22, p < .001) reduced this score by 39% to 1.11 points (Beta = 1.11, 95% CI = 0.39–1.83, p = .003).

## Discussion

We found, in a sample of people with and without HIV in South Africa, that estimated premorbid cognitive functioning is lower in people with HIV compared to people without HIV due, at least in part, to more adverse childhood psychosocial

**Table 2. Univariable associations with estimated premorbid cognitive performance and standard demographic, childhood psychosocial variables, current psychosocial variables and PCA components.**

| Variables | Beta | 95% CI | p |
|---|---|---|---|
| **Standard demographic** | | | |
| Sex (male vs. female) | 1.16 | 0.49–1.83 | .001 |
| Age at enrolment | −0.02 | −0.05–0.02 | .296 |
| Years of education | 0.66 | 0.45–0.87 | <.001 |
| **Childhood psychosocial** | | | |
| Childhood asset index (per 1 point higher) | 0.11 | 0.04–0.18 | .001 |
| Primary school quality of education (per quintile higher) | 0.55 | 0.20–0.90 | .002 |
| Secondary school quality of education (per quintile higher) | 0.73 | 0.43–1.03 | <.001 |
| Primary caregiver highest level of education (per year) | 0.36 | 0.14–0.58 | .001 |
| Primary caregiver occupation category (per 1 point higher) | −0.35 | −0.50−−0.20 | <.001 |
| Childhood trauma (CTQ score) | −0.00 | −0.02–0.02 | .988 |
| Exposure to violence (SECTV score) | 0.03 | −0.01–0.06 | .153 |
| **Current psychosocial** | | | |
| Occupation category | 0.09 | −0.07–0.25 | .276 |
| Depressive symptoms (CES-D > 16) | 0.68 | −0.14–1.49 | .102 |
| Monthly income from all sources (ZAR) | 0.04 | −0.10–0.17 | .592 |
| Current asset index (per 1 point higher) | 0.21 | 0.08–0.33 | .001 |
| Accommodation type | | | |
| Wendy house/backyard dwelling vs. shack | 0.31 | −0.43–1.05 | .406 |
| Own/family house vs. shack | 1.24 | 0.58–1.90 | <.001 |
| **PCA components** | | | |
| *Childhood Psychosocial Variables* component | 0.79 | 0.52–1.05 | <.001 |
| *Current Psychosocial Variables* component | 0.51 | −0.23–1.25 | .174 |
| *Experience of Childhood Trauma* component | 0.01 | −0.27–0.29 | .952 |

*Note*. 95% CI = 95% confidence interval.

**Table 3. Unadjusted HIV association with global cognitive performance, then adjusted for standard demographic variables, and estimated premorbid cognitive performance.**

| Variables | Model 1: Unadjusted Beta (95% CI) | p | Model 2: Adjustment for standard demographic variables Beta (95% CI) | p | Model 3: Adjustment for premorbid cognitive performance Beta (95% CI) | p | Model 4: Adjustment for premorbid cognitive performance and standard demographic variables Beta (95% CI) | p |
|---|---|---|---|---|---|---|---|---|
| HIV status | −3.95 (−5.48, −2.43) | <.001 | −2.92 (−4.34, −1.51) | <.001 | −2.70 (−4.19, −1.21) | <.001 | −2.23 (−3.62, −0.84) | .002 |
| Age (years) | − | | −0.17 (−0.25, −0.09) | <.001 | − | | −0.17 (−0.25, −0.09) | <.001 |
| Sex (male) | − | | −0.12 (−1.69, 1.45) | .881 | − | | −0.94 (−2.49, 0.61) | .234 |
| Years of education | − | | 1.56 (1.02, 2.09) | <.001 | − | | 1.13 (0.59, 1.67) | <.001 |
| WAIS-IV[SA] Information subtest | − | | − | | 0.92 (0.62, 1.22) | <.001 | 0.72 (0.42, 1.02) | <.001 |

circumstances (e.g., fewer household assets, poorer quality of education, parents with lower levels of educational achievement and occupational status). Of note is that our analyses observed these differences between people with and without HIV despite attempts to recruit sociodemographically similar cohorts (i.e., we recruited, from the same clinic, the

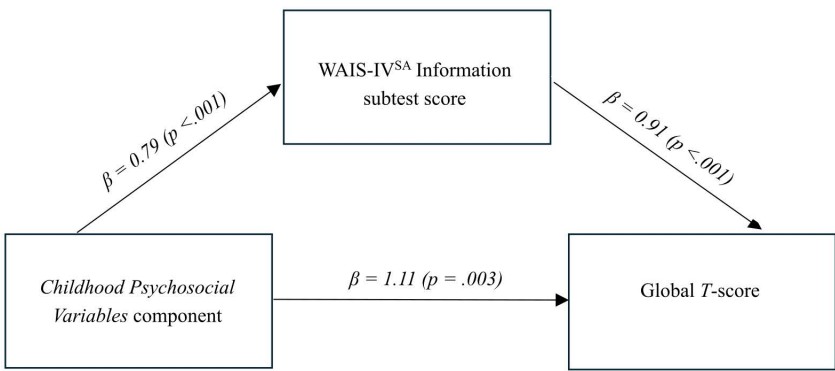

**Fig 1. Mediation analysis.**

friends, relatives, and acquaintances of people with HIV). We speculate that the presence of these residual background differences likely reflects psychosocial vulnerabilities to HIV acquisition in this region [14,30].

These data add to the previous findings of Dreyer et al. (2022), who described the effects of psychosocial confounds (which are rarely measured in neuro-HIV studies) on cognitive performance [12]. That paper contended that these psychosocial variables were likely exerting their effect on cognitive performance by differences in premorbid performance, i.e., that these people with HIV would have performed more poorly compared to people without HIV on cognitive tests even had they not contracted HIV. The current data support this speculation by demonstrating that people with HIV scored significantly lower than those without HIV on a cognitive test commonly used to estimate for premorbid cognitive performance.

These findings have important implications for the interpretation of cognitive test scores in people with HIV. Unless premorbid cognitive functioning is accounted for appropriately, low cognitive test performance in people with HIV could be misinterpreted as due to brain injury rather than to non-organic factors. This interpretation risks overestimating the degree of cognitive impairment caused by HIV-associated brain injury. The HAND criteria are particularly susceptible to this problem given that minimum criteria for HAND are met based on low cognitive test performance alone [1]. Overestimation of HIV-associated cognitive impairment has the potential to raise anxiety amongst people with HIV and contribute to stigma and discrimination towards them. False positive results could burden services and waste limited healthcare resources.

Controlling for WAIS-IV[SA] Information subtest reduced the effect of HIV infection on overall cognitive performance by 32%, which is a similar magnitude of decrease to what Dreyer et al. (2022) demonstrated when adjusting for numerous complex psychosocial factors [12]. This raised the possibility that adjusting for WAIS-IV[SA] Information subtest performance could be a simple way to adjust for the effect of complex psychosocial factors on cognitive performance. A similar approach using a different premorbid measure has previously been proposed [31]. In our cohort, although the magnitude was similar, the mediation analysis showed that the WAIS-IV[SA] Information subtest described somewhat different variability in the data as the psychosocial variables and did not fully mediate the relationship between psychosocial variables and global cognitive performance.

The lack of complete mediation could relate to a measurement error in our premorbid measure, a measurement error in our psychosocial measures, or both. The WAIS-IV[SA] Information subtest is a relatively crude measure of crystalised intelligence; the estimate of premorbid cognitive performance might be improved by adding other, more rigorous, tests (e.g., assessments of vocabulary and/or reading ability). Similarly, our psychosocial measures are likely to be crude as the constructs they seek to measure, i.e., the experience of poverty, discrimination, and lack of educational opportunity, are inherently complex and they are all self-report measures, and in some cases ask people to report on events/circumstances that

happened decades ago. Fundamentally, our analysis suggests that adjusting for premorbid cognitive differences using the WAIS-IV[SA] Information subtest does not fully mediate complex childhood psychosocial variables. As such, further research should examine this relationship in other cohorts before premorbid testing can be recommended as an approach to control for complex unmeasured psychosocial confounds.

In addition to creating bias in cognitive testing, lower estimates of premorbid cognitive performance in people with HIV may be indicative of the presence of lower levels of cognitive reserve. Hence, these individuals might have heightened vulnerability to cognitive impairment from HIV and other causes. This speculation is supported by data from a study of a United States cohort of people with HIV showing that only people with average premorbid functioning (compared to above-average) experience cognitive decline over time [32].

Our study has limitations. First, the sample was drawn from a low-income peri-urban South African population and hence our findings may not be generalisable to other cohorts of people with HIV, particularly those outside Africa where factors underlying HIV acquisition risk differ substantially. Second, all participants were receiving efavirenz-based antiretroviral therapy, which has been shown to affect cognitive performance and hence may have confounded our test results. In mitigation, however, is that these measures of premorbid cognitive functioning are intended to be resistant to these effects [7,16–18]. Third, we did not necessarily measure a complete set of childhood psychosocial variables that could impact premorbid cognitive performance. For example, we did not measure childhood food insecurity, a marker of extreme poverty that may have biological effects on cognition as a result of nutritional deficiencies or hunger [9]. Fourth, as discussed above, our estimate of premorbid cognitive functioning was relatively crude and accuracy could have been improved by adding additional tests.

In conclusion, testing of cognitive performance in people with HIV should be interpreted with caution to avoid overestimation of cognitive impairment caused by HIV, due to unaccounted for premorbid (i.e., pre-HIV acquisition) differences related to childhood psychosocial circumstances. We propose that estimates of premorbid cognitive performance, as well as psychosocial factors, be measured in studies investigating HIV effects on the brain (i.e., NeuroHIV studies).

## Acknowledgments

We would like to acknowledge the study team, especially Zimkhitha Ndinga and Teboho Linda, and the research study participant volunteers.

## Author contributions

**Conceptualization:** Anna J. Dreyer, Sam Nightingale.

**Data curation:** Anna J. Dreyer, Sam Nightingale.

**Formal analysis:** Anna J. Dreyer, Caroline Sabin.

**Funding acquisition:** Sam Nightingale.

**Investigation:** Anna J. Dreyer, John A. Joska, Sam Nightingale.

**Methodology:** Anna J. Dreyer, Caroline Sabin, Sam Nightingale.

**Project administration:** Anna J. Dreyer, Sam Nightingale.

**Resources:** John A. Joska, Sam Nightingale.

**Supervision:** John A. Joska, Kevin G. F. Thomas, Sam Nightingale.

**Writing – original draft:** Anna J. Dreyer, Sam Nightingale.

**Writing – review & editing:** Anna J. Dreyer, John A. Joska, Kevin G. F. Thomas, Caroline Sabin, Alan Winston, Saye Khoo, Sam Nightingale.

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
