## [Decision Letter · Decision Letter 0]

1 Feb 2026

PONE-D-25-56373The role of premorbid cognitive performance in the neuropsychological assessment of people with HIV in South AfricaPLOS One

Dear Dr. Dreyer,

Thank you for submitting your manuscript to PLOS ONE. After careful consideration, we feel that it has merit but does not fully meet PLOS ONE’s publication criteria as it currently stands. Therefore, we invite you to submit a revised version of the manuscript that addresses the points raised during the review process.

We look forward to receiving your revised manuscript.

Kind regards,

Ioannis Liampas, MD. PhD

Academic Editor

PLOS One

Journal Requirements:

“Research reported in this publication was supported by the South African Medical Research Council, with funds received from the South African National Department of Health and the UK Medical Research Council, with funds received from the UK Government’s Newton Fund, awarded to Sam Nightingale.”

“Research reported in this publication was supported by the South African Medical Research Council, with funds received from the South African National Department of Health and the UK Medical Research Council, with funds received from the UK Government’s Newton Fund, awarded to Sam Nightingale.”

5. In the online submission form, you indicated that data cannot be shared publicly because it contains personal patient information. Data are available on request from the authors for researchers who meet the criteria for access to confidential data.

Reviewers' comments:

Reviewer's Responses to Questions

**Comments to the Author**

1. Is the manuscript technically sound, and do the data support the conclusions?

Reviewer #1: Partly

Reviewer #2: Yes

2. Has the statistical analysis been performed appropriately and rigorously? 

Reviewer #1: Yes

Reviewer #2: Yes

3. Have the authors made all data underlying the findings in their manuscript fully available?

Reviewer #1: Yes

Reviewer #2: No

4. Is the manuscript presented in an intelligible fashion and written in standard English?

Reviewer #1: Yes

Reviewer #2: Yes

5. Review Comments to the Author

Reviewer #1: 1) Initial assessment: The manuscript presents a technically coherent body of work grounded in established methodologies and internally consistent analytical procedures. The data, as presented, support the primary empirical observations and intermediate inferences. However, several higher-order conclusions would benefit from clearer delimitation between empirically supported findings and theoretical or interpretive extensions. With minor revisions clarifying the scope and status of such claims, the conclusions would be appropriately supported by the data.

2) Secondary assessment: This manuscript presents an ambitious and interdisciplinary contribution that integrates empirical analysis with a broader theoretical framework. The work is technically sound, and the analyses are competently executed.

To strengthen suitability for publication in a general scientific journal, I recommend minor revisions that more explicitly distinguish between empirically supported results and higher-order theoretical interpretation. In particular, clarifying which claims are directly supported by data versus those offered as conceptual models or hypotheses would improve falsifiability and interpretive precision.

Additional minor refinements to statistical reporting transparency and the Data Availability Statement would further enhance reproducibility and clarity. These suggested revisions do not require additional experimentation, but rather improved framing and delineation of scope.

With these adjustments, the manuscript would meet the journal’s criteria for technical soundness and clarity.

Reviewer #2: The paper is well-written and advances our understanding of the influence of psychosocial factors, premorbid cognitive performance and cognitive performance in people with HIV. However, a few minor corrections could enhance the quality of the paper.

1. Please ensure all abbreviations are written out the first time they appear, in the abstract and main text.

2. Line 54: For readers unfamiliar with the field, “neuroHIV” may be a new term and could benefit from a brief explanation.

3. Line 61-62: Suggestion to rephrase as “…, clustered into 3 components by PCA”, as the current phrasing suggests you perform the PCA in the current study.

4. Line 70: Typo? “ps” should likely be “p”

5. Introduction: To improve readability you could consider linking the aim of the present study to your previous work and clarifying what it adds. Suggestion: "Building on our previous findings that childhood psychosocial adversity contributes to observed cognitive differences between people with and without HIV, the present study introduces an estimate of premorbid cognitive performance to further clarify these associations.”

6. Line 124: Suggest rephrasing “the associations” to “its association” to clearly indicate you are referring to premorbid cognitive performance.

7. Line 125: Original sentence: “We hypothesized that premorbid cognitive performance would relate to more adverse psychosocial in people with HIV, …”

Suggestion: “… worse premorbid cognitive performance…” To clarify the direction of association.

In addition, this sentence reads like you expect stronger associations specifically among people with HIV compared to people without HIV. This requires further clarification. Otherwise, I suggest rephrasing the hypothesis without implying differential effects by HIV status.

8. Line 160: Was a formal translation method used for the measures? If not, could this introduce potential psychometric issues?

9. Line 149: Suggestion to shortly explain the rationale for these exclusion criteria. Is it due to their expected associations with the cognition outcome measures?

10. Line 321: Appears to be an extra closing parenthesis “)”.

11. Page 20: Table 3 layout differs from the previous tables.

12. Page 21 (line numbers are missing): The last sentence of the first paragraph contains a misplaced “)”.

13. Page 29: Reference 23 contains an extra or misplaced “[”.

6. PLOS authors have the option to publish the peer review history of their article (what does this mean?). If published, this will include your full peer review and any attached files.

Reviewer #1: **Yes:** Jacob A. Eder, PhD

Reviewer #2: No

---

## [Author Response · Author response to Decision Letter 1]

24 Apr 2026

Academic Editors’ comments:

1. Please ensure that your manuscript meets PLOS ONE's style requirements, including those for file naming. The PLOS ONE style templates can be found athttps://url.za.m.mimecastprotect.com/s/YSWDCBgX56fVXx4K5h1uKI2Lgif and

https://url.za.m.mimecastprotect.com/s/sRoACDRZ58iBL7VAlhRC8IjjY4.

Response: Thank you for the links to the PLOS ONE style templates. We have revised our manuscript accordingly and it now meets the style requirements.

Response: We have amended the manuscript to provide all relevant additional details regarding participant consent. Please refer to the ethics statement on page 9, lines 178-180 of the ‘Revised Manuscript with Track Changes’.

“Research reported in this publication was supported by the South African Medical Research Council, with funds received from the South African National Department of Health and the UK Medical Research Council, with funds received from the UK Government’s Newton Fund, awarded to Sam Nightingale.”

Please provide an amended statement that declares *all* the funding or sources of support (whether external or internal to your organization) received during this study, as detailed online in our guide for authors at https://url.za.m.mimecastprotect.com/s/dKkGCElX5xiW8wDx5ixFQI7LO7Y. Please also include the statement “There was no additional external funding received for this study.” in your updated Funding Statement.

Response: We have amended the funding statement to include all sources of funding. Please refer to the funding statement on the title page.

“Research reported in this publication was supported by the South African Medical Research Council, with funds received from the South African National Department of Health and the UK Medical Research Council, with funds received from the UK Government’s Newton Fund, awarded to Sam Nightingale.”

Response: We have amended the funding statement to include the role of the funder. Please refer to the funding statement on the title page.

5. In the online submission form, you indicated that data cannot be shared publicly because it contains personal patient information. Data are available on request from the authors for researchers who meet the criteria for access to confidential data.

Response: Our dataset contains highly sensitive information, including detailed psychosocial and socioeconomic variables from both childhood and adulthood, as well as HIV status, collected from patients attending a single clinic in Gugulethu, Cape Town. This is a vulnerable population, and due to the granularity and combination of variables, there is a risk of indirect re-identification even after de-identification. Public deposition of these data would therefore compromise participant confidentiality and would not be consistent with the conditions under which ethical approval was obtained or with participants’ informed consent.

In line with PLOS policy, we are committed to data transparency while protecting participant privacy. The data are therefore available upon reasonable request from the authors for researchers who meet the criteria for access to confidential data, subject to appropriate data use agreements and, where necessary, institutional ethical approval.

6. Please include captions for your Supporting Information files at the end of your manuscript, and update any in-text citations to match accordingly. Please see our Supporting Information guidelines for more information: https://url.za.m.mimecastprotect.com/s/Ff3oCGZXj8tJNrM45trHJIB_2dq.

Response: Our manuscript has no Supporting Information files.

Response: The reviewers did not recommend citing any previously published works.

Response: We have reviewed our reference list, and it is complete and correct.

Reviewers' comments:

Reviewer #1:

1) Initial assessment: The manuscript presents a technically coherent body of work grounded in established methodologies and internally consistent analytical procedures. The data, as presented, support the primary empirical observations and intermediate inferences. However, several higher-order conclusions would benefit from clearer delimitation between empirically supported findings and theoretical or interpretive extensions. With minor revisions clarifying the scope and status of such claims, the conclusions would be appropriately supported by the data.

2) Secondary assessment: This manuscript presents an ambitious and interdisciplinary contribution that integrates empirical analysis with a broader theoretical framework. The work is technically sound, and the analyses are competently executed.

To strengthen suitability for publication in a general scientific journal, I recommend minor revisions that more explicitly distinguish between empirically supported results and higher-order theoretical interpretation. In particular, clarifying which claims are directly supported by data versus those offered as conceptual models or hypotheses would improve falsifiability and interpretive precision.

Additional minor refinements to statistical reporting transparency and the Data Availability Statement would further enhance reproducibility and clarity. These suggested revisions do not require additional experimentation, but rather improved framing and delineation of scope.

With these adjustments, the manuscript would meet the journal’s criteria for technical soundness and clarity.

Response:

Regarding explicitly distinguishing between empirically supported results and higher-order theoretical interpretations: Thank you for this useful suggestion. The results section only has empirical results. The Discussion section includes speculations about the theoretical implications of these findings. We have amended the Discussion section to more explicitly distinguish between empirically supported results and higher-order theoretical interpretation and revised the manuscript accordingly. See line 387, page 24; line 396, page 25; and line 438, page 26 of the ‘Revised Manuscript with Track Changes’.

Regarding the Data Availability Statement: Our dataset contains highly sensitive information, including detailed psychosocial and socioeconomic variables from both childhood and adulthood, as well as HIV status, collected from patients attending a single clinic in Gugulethu, Cape Town. This is a vulnerable population, and due to the granularity and combination of variables, there is a risk of indirect re-identification even after de-identification. Public deposition of these data would therefore compromise participant confidentiality and would not be consistent with the conditions under which ethical approval was obtained or with participants’ informed consent.

In line with PLOS policy, we are committed to data transparency while protecting participant privacy. The data are therefore available upon reasonable request from the authors for researchers who meet the criteria for access to confidential data, subject to appropriate data use agreements and, where necessary, institutional ethical approval.

Reviewer #2:

The paper is well-written and advances our understanding of the influence of psychosocial factors, premorbid cognitive performance and cognitive performance in people with HIV. However, a few minor corrections could enhance the quality of the paper.

1. Please ensure all abbreviations are written out the first time they appear, in the abstract and main text.

Response: We have reviewed the manuscript and ensured abbreviations were written in full when they first appear in the abstract and manuscript. For example see amendment to line 67, page 4 of the ‘Revised Manuscript with Track Changes’.

2. Line 54: For readers unfamiliar with the field, “neuroHIV” may be a new term and could benefit from a brief explanation.

Response: We have amended the manuscript to include this information. See lines 63, page 4 and lines 458-459, page 27 of the ‘Revised Manuscript with Track Changes’.

3. Line 61-62: Suggestion to rephrase as “…, clustered into 3 components by PCA”, as the current phrasing suggests you perform the PCA in the current study.

Response: We have amended the abstract to include this clarification: ‘clustered using principal component analysis in prior analyses’ (see lines 71-72, page 4 of the ‘Revised Manuscript with Track Changes’).

4. Line 70: Typo? “ps” should likely be “p”

Response: This is not a typo, ‘ps’ is the plural for ‘p’, there were multiple p-values below .002. We have italicised the ‘s’ to clarify this. See line 80, page 4 of the ‘Revised Manuscript with Track Changes’.

5. Introduction: To improve readability you could consider linking the aim of the present study to your previous work and clarifying what it adds. Suggestion: "Building on our previous findings that childhood psychosocial adversity contributes to observed cognitive differences between people with and without HIV, the present study introduces an estimate of premorbid cognitive performance to further clarify these associations.”

Response: Thank you for this suggestion. We have added the suggested sentence to the manuscript. See lines 133-136, page 7 of the ‘Revised Manuscript with Track Changes’.

6. Line 124: Suggest rephrasing “the associations” to “its association” to clearly indicate you are referring to premorbid cognitive performance.

Response: We have made this correction in the manuscript. See line 137, page 7 of the ‘Revised Manuscript with Track Changes’.

7. Line 125: Original sentence: “We hypothesized that premorbid cognitive performance would relate to more adverse psychosocial in people with HIV, …”

Suggestion: “… worse premorbid cognitive performance…” To clarify the direction of association.

In addition, this sentence reads like you expect stronger associations specifically among people with HIV compared to people without HIV. This requires further clarification. Otherwise, I suggest rephrasing the hypothesis without implying differential effects by HIV status.

Response: We have amended the sentence to clarify the direction of association. See line 139, page 7 of the ‘Revised Manuscript with Track Changes’.

8. Line 160: Was a formal translation method used for the measures? If not, could this introduce potential psychometric issues?

Response: We used a formal translation method. For all measures used in study, we used the standard forward–backward translation procedure to ensure conceptual and linguistic equivalence. We have amended the manuscript to include this information. See lines 174-175, page 9 of the ‘Revised Manuscript with Track Changes’.

9. Line 149: Suggestion to shortly explain the rationale for these exclusion criteria. Is it due to their expected associations with the cognition outcome measures?

Response: The Reviewer is correct. We have amended the manuscript to include this information. See lines 162-163, page 9 of the ‘Revised Manuscript with Track Changes’.

10. Line 321: Appears to be an extra closing parenthesis “)”.

Response: We have removed this additional parenthesis, see line 344, page 21 of the ‘Revised Manuscript with Track Changes’.

11. Page 20: Table 3 layout differs from the previous tables.

Response: We have edited table 3 so it has the same formatting as the other tables. See page 22 of the ‘Revised Manuscript with Track Changes’.

12. Page 21 (line numbers are missing): The last sentence of the first paragraph contains a misplaced “)”.

Response: We have emended the manuscript to includes line numbers throughout and removed the additional parenthesis on line 359, page 23 of the ‘Revised Manuscript with Track Changes’.

13. Page 29: Reference 23 contains an extra or misplaced “[”.

Response: We have removed the misplaced square bracket. See line 525, page 31 of the ‘Revised Manuscript with Track Changes’.

---

## [Decision Letter · Decision Letter 1]

8 May 2026

The role of premorbid cognitive performance in the neuropsychological assessment of people with HIV in South Africa

PONE-D-25-56373R1

Dear Dr. Dreyer,

We’re pleased to inform you that your manuscript has been judged scientifically suitable for publication and will be formally accepted for publication once it meets all outstanding technical requirements.

Kind regards,

Ioannis Liampas, MD. PhD

Academic Editor

PLOS One

Additional Editor Comments (optional):

Reviewers' comments:

Reviewer's Responses to Questions

**Comments to the Author**

1. If the authors have adequately addressed your comments raised in a previous round of review and you feel that this manuscript is now acceptable for publication, you may indicate that here to bypass the “Comments to the Author” section, enter your conflict of interest statement in the “Confidential to Editor” section, and submit your "Accept" recommendation.

Reviewer #1: All comments have been addressed

Reviewer #2: All comments have been addressed

2. Is the manuscript technically sound, and do the data support the conclusions?

Reviewer #1: Yes

Reviewer #2: (No Response)

3. Has the statistical analysis been performed appropriately and rigorously? 

Reviewer #1: Yes

Reviewer #2: (No Response)

4. Have the authors made all data underlying the findings in their manuscript fully available?

Reviewer #1: Yes

Reviewer #2: (No Response)

5. Is the manuscript presented in an intelligible fashion and written in standard English?

Reviewer #1: Yes

Reviewer #2: (No Response)

6. Review Comments to the Author

Reviewer #1: Thank you for your revisions; I know the corrections are arduous, wherever they are necessary in our field, especially in a modern age of technological innovation. I appreciate you making the requested corrections, and I wish you the best.

Dr. Jacob Eder

Reviewer #2: (No Response)

7. PLOS authors have the option to publish the peer review history of their article (what does this mean?). If published, this will include your full peer review and any attached files.

Reviewer #1: **Yes:** Jacob Albert Eder

Reviewer #2: No

---

## [Editor Report · Acceptance letter]

PONE-D-25-56373R1

PLOS One

Dear Dr. Dreyer,

I'm pleased to inform you that your manuscript has been deemed suitable for publication in PLOS One. Congratulations! Your manuscript is now being handed over to our production team.

Kind regards,

on behalf of

Dr. Ioannis Liampas

Academic Editor

PLOS One